

# Mechanisms of NLRP3 inflammasome in pathogenesis and progression of inflammation-related gastrointestinal diseases

Fengmei Liu[1,*], Bozong Shao[2,*], Yaqin Zhu[3], Xiaochun Xue[4] and Xiaoyan Wu[5]

[1] Department of Nursing, 905th Hospital of People's Liberation Army Navy, Shanghai, China
[2] Department of Gastroenterology, Chinese PLA General Hospital, Beijing, China
[3] Center of Digestive Endoscopy, 905th Hospital of People's Liberation Army Navy, Shanghai, China
[4] Department of Pharmacy, 905th Hospital of People's Liberation Army Navy, Shanghai, China
[5] Department of Gastroenterology, 905th Hospital of People's Liberation Army Navy, Shanghai, China
[*] These authors contributed equally to this work.

Corresponding authors
Xiaochun Xue, xxc2021@126.com
Xiaoyan Wu, 16565667@qq.com

## ABSTRACT

The inflammasome is a novel component of the innate immune response. It plays a crucial role in the pathogenesis and progression of inflammation-related gastrointestinal diseases. Among various inflammasomes, the NLR family pyrin domain containing 3 (NLRP3) inflammasome is one of the most extensively studied. Increasingly, researchers are exploring its roles and mechanisms, particularly in inflammation-immune-related diseases. As a result, a review paper is demanded to review and summarize the previous and latest studies on the role and mechanisms of NLRP3 inflammasome in pathogenesis and progression of inflammation-related gastrointestinal diseases. This review comprehensively elaborates on the biological characteristics of the NLRP3 inflammasome, including its assembly and activation mechanisms. Additionally, it emphasizes the roles and mechanisms of the NLRP3 inflammasome in common inflammation-related gastrointestinal diseases such as ulcerative colitis, Crohn's disease, pancreatitis, and non-alcoholic fatty liver disease (NAFLD). Furthermore, the application of NLRP3 inflammasome inhibitors in treating these diseases is discussed. Articles from PubMed and Web of Science on NLRP3 inflammasome, ulcerative colitis, Crohn's disease, pancreatitis, and NAFLD were summarized to analyze the data and conclusions carefully to ensure the comprehensiveness, completeness, and accuracy of the review. This study aims to provide scholars engaged in research on gastrointestinal diseases with new directions for developing more effective therapeutics for inflammation-related gastrointestinal diseases by investigating the NLRP3 inflammasome's role in these conditions.

## INTRODUCTION

Inflammation-related gastrointestinal diseases are a group of disorders characterized by inflammation in the digestive system (*Yamamoto, Kawada & Obama, 2021*; *Zhao et al.,*

---

2023a). These diseases can be classified into acute and chronic inflammatory conditions based on their onset and progression. During disease development, pathogenic factors can excessively activate inflammatory immune responses, recruiting a large number of immune cells to infiltrate digestive organs, leading to localized or systemic inflammatory cascades and tissue damage (*He et al., 2021*; *Li et al., 2023a*; *Yang & Yuan, 2023*).

Innate immune responses act as a critical defense barrier against endogenous and exogenous threats by recognizing antigens through pattern-recognition receptors (PRRs) and presenting them to immune cells such as macrophages, thereby activating downstream immune and inflammatory responses (*Carroll, Pasare & Barton, 2024*; *Duan et al., 2022*; *Li et al., 2023c*; *Lu et al., 2024*). However, excessive activation of innate immune responses may contribute to the onset and progression of various inflammation-related diseases (*Carty, Guy & Bowie, 2021*; *Castro-Gomez & Heneka, 2024*).

This review aims to summarize the biological characteristics of the NLRP3 inflammasome, a key component of innate immune responses, and delve into its role in several common inflammation-related gastrointestinal diseases, such as ulcerative colitis, Crohn's disease, pancreatitis, and non-alcoholic fatty liver disease (NAFLD). By integrating these findings, the review seeks to provide a theoretical foundation and research direction for developing novel therapeutic strategies for these diseases. We believe that this review will offer new insights into uncovering the mechanisms of inflammation-related gastrointestinal diseases and advancing the development of treatments for such conditions.

## SURVEY METHODOLOGY

This review analyzed relevant literature published between 2002 and 2024, retrieved from PubMed (https://pubmed.ncbi.nlm.nih.gov/) and Web of Science (https://www.webofscience.com/). The search was conducted by combining subject terms and free-text words. The following heading terms were used: "NLRP3 inflammasome", "gastrointestinal diseases", "inflammatory bowel disease", "ulcerative colitis", "Crohn's disease", "pancreatitis", "non-alcoholic fatty liver disease", and "inflammation". An initial screening of the literature titles was performed, followed by a secondary screening of abstracts and keywords. Finally, the full texts were obtained for further evaluation.

### Biological characteristics of NLRP3 inflammasome

Over the past few decades, extensive research has been conducted to elucidate the characteristics of the NLRP3 inflammasome and its roles in various diseases (*Olona, Leishman & Anand, 2022*; *Shao et al., 2014*; *Zahid et al., 2019*). This section introduces the general characteristics of the NLRP3 inflammasome, followed by an in-depth discussion of its activation mechanisms and its involvement in various pathological conditions.

### *General information of NLRP3 inflammasome*

The innate immune response serves as a critical defense mechanism in mammals against both internal and external threats (*Li et al., 2023b*; *Ma et al., 2023*). It relies on PRRs to identify antigens, which are then presented to inflammatory immune cells, such as macrophages, triggering downstream immune and inflammatory responses (*House et*

*al., 2023*; *Tang et al., 2012*; *Zhao et al., 2023b*). Pathogen-associated molecular patterns (PAMPs), commonly recognized by innate immunity, are detected by receptors such as toll-like receptors (TLRs), C-type lectins (CTLs), and galectins (*Fernandes-Santos & Azeredo, 2022*; *Fitzgerald & Kagan, 2020*; *Kawai et al., 2024*; *Peters & Peters, 2021*).

Inflammasomes, first discovered and characterized in 2002, are a crucial component of the immune response. As part of the innate immune system, they have been extensively reported to be closely associated with various immune and inflammatory pathways. These include nuclear factor kappa B (NF-κB) signaling, mitogen-activated protein kinase (MAPK) signaling, and Janus kinase (JAK)-signal transducer and activator of transcription (STAT) signaling (*Chen et al., 2023b*; *De Gaetano et al., 2021*; *Ma, 2023*; *Vande Walle & Lamkanfi, 2024*; *Vervaeke & Lamkanfi, 2025*). To date, various inflammasomes have been identified, including NLRP1, NLRP2, NLRP3, AIM2, and NLRC4 (*Chen et al., 2023a*; *Shao et al., 2023*; *Xu et al., 2024*). Among these, the NLRP3 inflammasome is the most extensively studied. Current research highlights its pivotal role in the onset and progression of numerous diseases, warranting a detailed examination of its biological characteristics and pathogenic roles in this review.

The NLRP3 inflammasome is a multiprotein complex comprising three components: the NLRP3 protein, procaspase-1, and the adaptor protein apoptosis-associated speck-like protein containing a CARD (ASC) (*Zhan et al., 2022*; *Zhang et al., 2021b*). Under normal conditions, the NACHT domain and leucine-rich repeats (LRRs) of NLRP3 remain tightly bound, preventing its interaction with ASC and subsequent inflammasome assembly. However, upon stimulation by PAMPs or damage-associated molecular patterns (DAMPs), the conformation of the NLRP3 protein changes, allowing the pyrin domains (PYDs) of NLRP3 to interact with the corresponding domains of ASC. This interaction facilitates the recruitment of procaspase-1, whose CARD domain binds to the corresponding domain of ASC, culminating in the assembly of the NLRP3 inflammasome and the initiation of downstream inflammatory cascades (*Luo et al., 2023*; *Park et al., 2023*; *Shao et al., 2015*; *Zheng et al., 2023*).

### Activation of NLRP3 inflammasome

According to previous reviews from us and other researchers, the activation of the NLRP3 inflammasome occurs in two primary steps (*Fu & Wu, 2023*; *Huang, Xu & Zhou, 2021*; *Shao, Cao & Liu, 2018*; *Shao et al., 2019b*).

1. **Priming step:** In response to various exogenous and endogenous PAMPs or DAMPs, PRRs such as TLRs are activated, triggering nuclear factor kappa B (NF-κB)-mediated signaling pathways. This signaling promotes the transcription and synthesis of inflammasome-related proteins, including NLRP3, pro-interleukin (IL)-1β, and pro-IL-18, preparing the system for subsequent activation.

2. **Activation step:** Further stimulation induces NLRP3 oligomerization and the recruitment of ASC and procaspase-1. These components interact through their respective domains, forming the NLRP3 inflammasome complex. The formation of this complex converts procaspase-1 into the active enzyme caspase-1, which then processes pro-IL-1β and pro-IL-18 into their mature forms, IL-1β and IL-18. These pro-inflammatory cytokines

are subsequently released extracellularly, initiating localized or systemic inflammatory responses.

Multiple stimuli have been identified as activators of the NLRP3 inflammasome. For the priming step, the Gram-negative bacterial outer membrane component lipopolysaccharide (LPS) activates TLR4, enhancing the transcription of NLRP3-related proteins and facilitating inflammasome assembly (*Unamuno et al., 2021*; *Zhang et al., 2021a*). During the activation step, diverse PAMPs and DAMPs have been implicated, including adenosine triphosphate (ATP), β-amyloid, silica, reactive oxygen species (ROS), asbestos, cathepsin B, and mitochondrial $Ca^{2+}$ overload (*Liu et al., 2022a*; *Liu et al., 2022b*; *Xian et al., 2022*).

While physiological activation of the NLRP3 inflammasome is essential for host defense, its dysregulated or excessive activation contributes to the pathogenesis of numerous diseases, including cardiovascular conditions (*e.g.*, myocardial infarction, atherosclerosis), respiratory diseases (*e.g.*, pneumonia, tuberculosis), gastrointestinal disorders (*e.g.*, inflammatory bowel disease, pancreatitis), metabolic diseases (*e.g.*, obesity, diabetes), and malignancies (*Olona, Leishman & Anand, 2022*; *Paik et al., 2021*; *Seoane et al., 2020*; *Toldo & Abbate, 2024*; *Toldo et al., 2022*). Increasing attention has been directed towards modulating NLRP3 inflammasome activity as a therapeutic approach, particularly in gastrointestinal diseases (*Arre et al., 2023*; *Donovan et al., 2020*; *Nani & Tehami, 2023*). Subsequent sections will explore the roles and mechanisms of the NLRP3 inflammasome in inflammation-related gastrointestinal diseases such as ulcerative colitis, Crohn's disease, pancreatitis, and NAFLD.

## NLRP3 inflammasome in inflammation-related gastrointestinal diseases

As a key component of the innate immune response, the NLRP3 inflammasome plays a pivotal role in the pathogenesis and progression of inflammation-related diseases. In this section, we discuss its roles and underlying mechanisms in several common inflammation-related gastrointestinal diseases, including ulcerative colitis, Crohn's disease, pancreatitis, and NAFLD (shown in Fig. 1 and Table 1).

### NLRP3 inflammasome in ulcerative colitis

Ulcerative colitis is a chronic, nonspecific inflammatory bowel disease characterized by mucosal and submucosal inflammation, primarily affecting the rectum and distal colon (*Le Berre, Honap & Peyrin-Biroulet, 2023*). Patients often present with intestinal bleeding, mucus-laden stools accompanied by tenesmus, and lower abdominal pain. Ulcerative colitis is associated with a significantly higher risk of malignancy compared to the general population. While its etiology remains unclear, potential contributors include genetic predisposition, immune dysregulation, environmental factors, and infections (*Riviere et al., 2024*). Disruption of the intestinal mucosal barrier by pathogens or excessive immune responses triggered by genetic susceptibility may damage the intestinal lining, promoting inflammation. Macrophage activation plays a crucial role in this process (*Yang et al., 2022*; *Zhang et al., 2020*; *Zhang et al., 2023b*). LPS from bacteria or necrotic intestinal tissues can recruit and activate macrophages *via* TLRs, leading to the release of pro-inflammatory

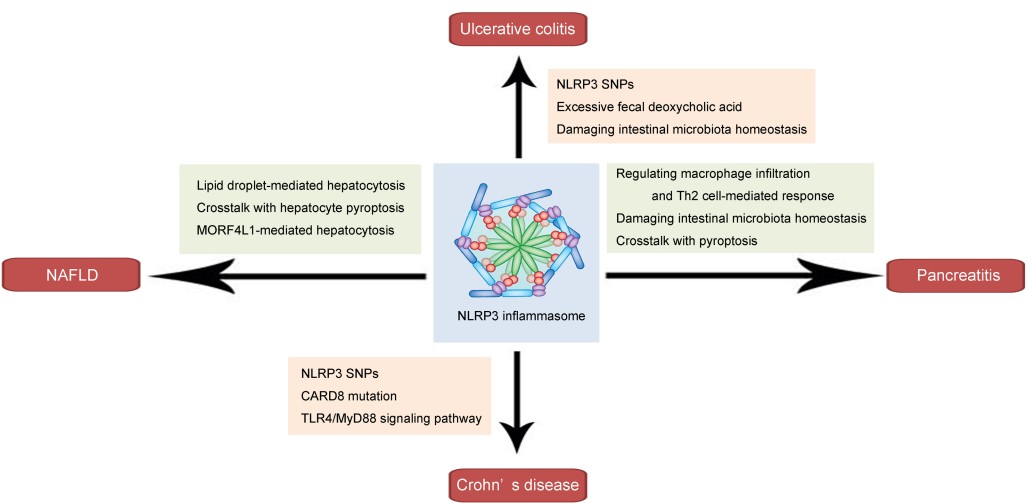

**Figure 1** **Schematic illustration of the role and mechanism of the NLRP3 inflammasome in inflammation-related gastrointestinal diseases.** The NLRP3 inflammasome affects ulcerative colitis *via* NLRP3 SNPs, involving excessive fecal deoxycholic acid-mediated intestinal damage, and damaging intestinal microbiota homeostasis. The NLRP3 inflammasome influences Crohn's disease through NLRP3 SNPs, CARD8 mutation and TLR4/MyD88 signaling pathway. In pancreatitis, the NLRP3 inflammasome may regulate macrophage infiltration and Th2 cell-mediated response, damage intestinal microbiota homeostasis, and interact with pyroptosis. The NLRP3 inflammasome affects NAFLD through involving lipid droplet-mediated hepatocytosis, crosstalk with hepatocyte pyroptosis, and MORF4L1-mediated hepatocytosis.

cytokines and exacerbation of ulcerative colitis (*Burdette et al., 2021*; *Candelli et al., 2021*; *Larabi, Barnich & Nguyen, 2020*).

Studies have shown that NLRP3 inflammasome-mediated release of IL-1β and IL-18 from macrophages aggravates inflammation in ulcerative colitis, further impairing the intestinal defense system and driving disease progression (*Ali et al., 2023*; *Zaharie et al., 2023*; *Zhang et al., 2023a*). Clinical analyses comparing healthy individuals and ulcerative colitis patients revealed that NLRP3 single nucleotide polymorphisms (SNPs), such as rs10754558 and rs10925019, are significantly associated with ulcerative colitis susceptibility (*Hanaei et al., 2018*; *Zhang et al., 2014*). Additionally, high-fat diet-induced excessive fecal deoxycholic acid (DCA), an endogenous DAMP, has been implicated in NLRP3 inflammasome activation and DSS-induced colitis (*Zhao et al., 2018*). Dysregulation of intestinal microbiota, linked to NLRP3 inflammasome overactivation, has also been observed, with elevated levels of pathogenic *Escherichia coli* and *Lactobacillus* strongly correlating with disease severity (*Zhang et al., 2017a*; *Zhang et al., 2017b*).

Therapeutic interventions targeting NLRP3 inflammasome have demonstrated efficacy in ulcerative colitis models. Our previous studies showed that α7 nicotinic acetylcholine receptor knockout exacerbated dextran sulfate sodium (DSS)-induced colitis in mice, increasing systemic inflammation and intestinal infiltration. This receptor suppresses NLRP3 inflammasome activation *via* autophagy, mitigating ulcerative colitis severity (*Shao*

Liu et al. (2025), *PeerJ*, DOI 10.7717/peerj.19828

**Table 1** Potential pharmacological mechanisms of NLRP3 inflammasome inhibitors in inflammation-related gastrointestinal disease treatment.

| Inflammation-related gastrointestinal disease | NLRP3 inflammasome inhibitor | Potential pharmacological mechanisms | References |
|---|---|---|---|
| Ulcerative colitis | Alpha7 nicotinic acetylcholine receptor | Upregulating autophagy process | *Shao et al. (2019a)* |
| | Cannabinoid receptor 2 | Inducing AMPK-mTOR-p70S6K signaling pathway | *Ke et al. (2016)* |
| | Gentianine | Inhibiting TLR4/NLRP3-mediated pyroptosis | *Li et al. (2024)* |
| Crohn's disease | Polyphenolic extract rich in anthocyanins | Suppressing mast cells activation | *Ortiz-Cerda et al. (2023)* |
| Pancreatitis | High-density lipoprotein | Inhibiting acinar cell pyroptosis | *Lu et al. (2023)* |
| | Naringenin | Increasing AhR nuclear translocation and activating the AhR pathway | *Yan et al. (2023)* |
| | Baicalein | MiR-192-5p upregulation and TXNIP inhibition | *Wang et al. (2021)* |
| Non-alcoholic fatty liver disease | MCC950 | Inhibiting NLRP3/Caspase-1/IL-1β and NF-κB/NLRP3 inflammasome signaling pathway | *Mridha et al. (2017)*; *Qi et al. (2023)*; *Zou, Yan & Wan (2022)* |
| | Lycopene | Inhibiting NF-κB/NLRP3 inflammasome signaling pathway | *Gao et al. (2023)* |
| | Echinatin | Regulating the combination of NLRP3 inflammasome and heat-shock protein 90 | *Xu et al. (2021)* |

*et al., 2019a*). Similarly, activation of cannabinoid receptor 2 (CB2R) alleviated DSS-induced colitis by inhibiting the adenosine 5-monophosphate (AMP)-activated protein kinase (AMPK)-mammalian target of rapamycin rabbit (mTOR)-p70 ribosomal protein S6 kinase (p70S6K) pathway dependent autophagic regulation of NLRP3 inflammasome (*Ke et al., 2016*). Furthermore, Acupuncture at the Juxu acupoint can regulate the abnormal expression of IL-1β, thereby improving intestinal mucosal damage. Gentianine, a traditional herbal compound, significantly reduced colitis severity through TLR4/NLRP3-mediated pyroptosis inhibition (*Li et al., 2024*). Despite extensive investigation into the role of the NLRP3 inflammasome in the pathogenesis and progression of ulcerative colitis, few therapeutic agents targeting its suppression have successfully transitioned to clinical practice. Consequently, further research and development of such drugs are required for the treatment of ulcerative colitis.

### NLRP3 inflammasome in Crohn's disease

Crohn's disease is a chronic granulomatous inflammatory disorder, most commonly affecting the terminal ileum and adjacent colon (*Torres et al., 2017*). It is characterized by digestive ulcers, intestinal strictures, and perforations, with high recurrence rates (*Dolinger, Torres & Vermeire, 2024*). Symptoms include abdominal pain, diarrhea, and weight loss, and the etiology is thought to involve genetic, environmental, microbiota, and immune factors (*Massironi et al., 2023*). First-line treatments include aminosalicylates, corticosteroids, immunosuppressants, and biologics such as infliximab. However, therapeutic outcomes remain suboptimal due to an incomplete understanding of disease mechanisms.

Multiple studies have demonstrated a strong association between NLRP3 inflammasome and Crohn's disease. Analyses of European populations revealed a significant link between NLRP3 SNPs and Crohn's disease susceptibility (*Villani et al., 2009*). Additionally, it has been reported that in Crohn's disease patients, the presence of the major alleles of NLRP3 SNPs rs10733113, rs55646866, and rs4353135 correlates with fewer surgeries and a lower maximal Crohn's Disease Activity Index (CDAI) (*Yoganathan et al., 2021*). Loss-of-function mutations in CARD8, a negative inflammasome regulator, were also significantly associated with Crohn's disease, underscoring its critical role in disease progression (*Mao et al., 2018*; *Vasseur et al., 2013*). In 2,4,6-trinitrobenzene sulphonic acid (TNBS)-induced Crohn's disease mouse models, inhibition of TLR4/MyD88 signaling effectively reduced NLRP3 inflammasome activation and ameliorated disease severity (*Luo et al., 2017*).

Research into therapeutic targeting of NLRP3 inflammasome in Crohn's disease has yielded promising results. Polyphenolic extracts rich in anthocyanins from *Maqui* inhibited NLRP3 inflammasome activation and mast cell activity in TNBS-induced Crohn's disease-like colitis models, demonstrating anti-inflammatory effects at various disease stages (*Ortiz-Cerda et al., 2023*). However, despite these exciting findings, translating them into first-line clinical therapies for Crohn's disease remains a critical challenge that requires further investigation.

### NLRP3 inflammasome in pancreatitis

Pancreatitis is a non-infectious inflammatory disease broadly classified into acute pancreatitis and chronic pancreatitis (*Mitchell, Byrne & Baillie, 2003*). Acute pancreatitis is characterized by acute injuries such as edema, hemorrhage, and necrosis of pancreatic tissue caused by various factors (*Boxhoorn et al., 2020*). In contrast, chronic pancreatitis involves chronic, progressive inflammation leading to irreversible damage to pancreatic exocrine and endocrine functions (*Beyer et al., 2020*). Major symptoms include acute abdominal pain, nausea, vomiting, fever, and in severe cases, acute multi-organ dysfunction. The primary etiologies are biliary diseases, such as gallstones and biliary infections, and alcohol-induced damage (*Mayerle et al. , 2019*). The disease pathogenesis primarily involves activation of digestive enzymes stored in the pancreas, triggering acute inflammatory storms or chronic recurrent immune responses that result in pancreatic tissue injury and autodigestion (*Wood et al., 2020*). Current therapeutic strategies include nutritional support, fluid resuscitation, antibiotic prophylaxis, and somatostatin analogs to suppress pancreatic exocrine secretion, though their effectiveness remains limited.

Emerging research highlights the significant role of the NLRP3 inflammasome in pancreatitis progression. A study in animal models of acute pancreatitis demonstrated that specific deletion of NLRP3 exacerbates disease severity by disrupting gut microbiota homeostasis (*Li et al., 2020*), underscoring the link between NLRP3 inflammasome and pancreatitis. Another study revealed that in mice with severe pancreatitis, NLRP3 inhibition ameliorates disease severity by regulating macrophage infiltration and Th2 cell-mediated responses *via* IL-18, thereby reducing systemic inflammatory response syndrome (SIRS) and compensatory anti-inflammatory response syndrome (CARS) (*Sendler et al., 2020*). Furthermore, NLRP3 inflammasome activation is intricately associated with pyroptosis, where its overactivation in pancreatitis promotes GSDMs-mediated programmed cell death, exacerbating pancreatic tissue injury (*Al Mamun et al., 2022*).

Recent studies have elucidated therapeutic strategies targeting NLRP3 inflammasome in pancreatitis. Previous reports indicate a positive correlation between serum high-density lipoprotein (HDL) levels and the severity of acute pancreatitis (*Ni et al., 2023*; *Yang, He & Wang, 2024*). HDL has been shown to protect against acinar cell death both *in vivo* and *in vitro* by inhibiting NLRP3 inflammasome signaling and acinar cell pyroptosis (*Lu et al., 2023*). Additionally, the natural compound naringenin mitigates acute pancreatitis-associated intestinal injury by suppressing NLRP3 inflammasome activation through increased AhR nuclear translocation and AhR pathway activation (*Yan et al., 2023*). Moreover, baicalein, a traditional Chinese herbal component, exhibits protective effects in hyperlipidemic pancreatitis by inhibiting NLRP3 inflammasome activation through miR-192-5p upregulation and TXNIP inhibition (*Wang et al., 2021*). Several agents have demonstrated therapeutic effects in treating pancreatitis by inhibiting the NLRP3 inflammasome in preclinical research. However, the clinical application of NLRP3 inflammasome inhibitors in this context remains under-explored and warrants further investigation.

### NLRP3 inflammasome in NAFLD

NAFLD encompasses a spectrum of liver pathologies characterized by excessive lipid deposition in hepatocytes in the absence of alcohol or other identifiable liver-damaging factors (*Pouwels et al., 2022*). NAFLD includes simple steatosis (SS), non-alcoholic steatohepatitis (NASH), and related cirrhosis (*Paternostro & Trauner, 2022*). It is closely linked to insulin resistance and genetic predisposition. The hallmark pathological changes involve hepatic lipid degeneration and inflammation, manifesting as inflammatory infiltration and hepatocyte injury (*Maurice & Manousou, 2018*). Dysregulated inflammatory immune cascades play a pivotal role in NAFLD progression, and suppressing excessive inflammatory responses represents a promising therapeutic strategy.

Research indicates that NLRP3 inflammasome activation significantly promotes NAFLD development and progression (*De Carvalho Ribeiro & Szabo, 2022*; *Wong et al., 2023*; *Yu et al., 2022*). Previous studies have shown that the NLRP3 inflammasome mediates hepatocyte damage *via* lipid droplet (LD)-dependent mechanisms, disrupting hepatic microenvironment homeostasis through the LD-membrane-spanning 4-domains subfamily A member 7 (MS4A7)-NLRP3 axis (*Zhou et al., 2024*). Additionally, the interaction between NLRP3 inflammasome and hepatocyte pyroptosis further exacerbates hepatocyte injury and death (*Gaul et al., 2021*). Another study revealed that the NLRP3 inflammasome participates in hepatocyte damage mediated by mortality factor 4-like protein 1 (MORF4L1), promoting NAFLD progression through the mitochondrial MORF4L1-TUFM regulatory pathway (*Tian et al., 2022*).

Recent studies highlight the therapeutic potential of NLRP3 inflammasome inhibition in NAFLD. For instance, the NLRP3 inhibitor MCC950 significantly suppresses liver inflammation and fibrosis in animal models and NAFLD patients, demonstrating efficacy through modulation of the NLRP3/Caspase-1/IL-1β and NF-κB/NLRP3 inflammasome pathways (*Mridha et al., 2017*; *Qi et al., 2023*; *Zou, Yan & Wan, 2022*). Furthermore, lycopene, a natural compound extracted from red fruits and vegetables, mitigates NAFLD progression by inhibiting the NF-κB/NLRP3 inflammasome pathway (*Gao et al., 2023*). Additionally, echinatin, an active ingredient in licorice, effectively suppresses NLRP3 inflammasome activation, exerting hepatoprotective and anti-NAFLD effects *via* regulation of the NLRP3-HSP90 interaction (*Xu et al., 2021*). To date, several agents have demonstrated efficacy in alleviating NAFLD, but none have yet been successfully translated into clinical practice. The development and clinical application of novel drugs targeting NLRP3 inflammasome inhibition for NAFLD treatment remain areas where significant progress is still required.

## CONCLUSIONS

In summary, this study comprehensively reviews the biological characteristics of the NLRP3 inflammasome, with a focus on its roles and mechanisms in several inflammation-related gastrointestinal diseases, including ulcerative colitis, Crohn's disease, pancreatitis, and NAFLD. Recent advancements in this field have led to the identification of various NLRP3

inflammasome inhibitors with significant therapeutic potential for these conditions. However, the clinical application of these inhibitors remains limited. Further research is needed to develop effective NLRP3 inflammasome-targeted therapeutic agents for managing inflammation-related gastrointestinal diseases. Future work should focus on exploring the specific mechanisms of these diseases in relation to inflammatory and immune responses, and identifying more potential and effective NLRP3 inflammasome inhibitors for treating such conditions.

### Funding
The authors received no funding for this work.

### Competing Interests
The authors declare there are no competing interests.

### Author Contributions
- Fengmei Liu conceived and designed the experiments, performed the experiments, analyzed the data, prepared figures and/or tables, authored or reviewed drafts of the article, and approved the final draft.
- Bozong Shao conceived and designed the experiments, prepared figures and/or tables, authored or reviewed drafts of the article, and approved the final draft.
- Yaqin Zhu performed the experiments, analyzed the data, prepared figures and/or tables, and approved the final draft.
- Xiaochun Xue conceived and designed the experiments, prepared figures and/or tables, authored or reviewed drafts of the article, and approved the final draft.
- Xiaoyan Wu conceived and designed the experiments, prepared figures and/or tables, authored or reviewed drafts of the article, and approved the final draft.

### Data Availability
  This is a literature review.

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
