# Peer review of "Mechanisms of NLRP3 inflammasome in pathogenesis and progression of inflammation-related gastrointestinal diseases"

_PeerJ, doi:10.7717/peerj.19828_

## Round 0.1 · original submission · Minor Revisions

Please address the comments of all reviewers and amend the manuscript accordingly.

Reviewer 1 ·

Basic reporting

The language used throughout the paper is clear and professional. The introduction effectively establishes the context of inflammation-related gastrointestinal diseases and the role of the innate immune system, specifically focusing on the NLRP3 inflammasome. The literature cited is relevant and provides a good overview of the current understanding of the NLRP3 inflammasome and its involvement in various diseases.
However, the importance of this review must be emphasized further at the end of the introduction.

1 - Some redundancy is present in the introduction and the General characteristics of NLRP3 inflammasome; consider deleting the latter paragraph or adding different information.
The redundant paragraph
“Inflammasomes, a novel component of the innate immune response, were first identified in 2002 as a bridge between inflammation and immune responses (Fu & Wu 2023; Rathinam & Fitzgerald 2016). Several inflammasomes, including NLR family pyrin domain containing 1 (NLRP1), NLRP2, NLRP3, absent in melanoma 2 (AIM2), and NLRC4, have been characterized, with the NLRP3 inflammasome being the most extensively studied (Chen et al. 2023a; Shao et al.2023; Xu et al. 2024).”

Also
2 - The title “PART I: BIOLOGICAL CHARACTERISITICS OF NLRP3 INFLAMMASOME” is very similar to the subtitle “General characteristics of NLRP3 inflammasome.” Consider combining them.

3 - Since the inflammasome was first discovered in 2002, it is no longer novel to replace this word with alternatives

4 - No need to add part 1 and part 2 in the title, perhaps using numbering is better: 1. 2., and 1.1. 1.2 etc…

Experimental design

1 - Although databases are mentioned, there is a lack of a clearly defined search strategy; the specific search strategy (e.g., inclusion/exclusion criteria, date range) could be more detailed.

2 - Simply stating keywords were used is insufficient for reproducibility. Analyzing data and conclusions "carefully" to ensure comprehensiveness, lacks specific details on the process.

Validity of the findings

1 - A review should not only summarize existing knowledge but also critically evaluate its significance and identify gaps in the research. Therefore, impact and novelty must be mentioned.

2 - The conclusion mentions the need for further research, but it lacks specific suggestions for future studies. Identifying key areas for investigation would enhance the value of the review.

3 - Additionally, the review mentions potential pharmacological mechanisms of NLRP3 inhibitors (Table 1) but lacks source citations. Adding a "References" column to Table 1, citing the specific studies that support each mechanism, is crucial for transparency and allows readers to critically assess the treatments.

Reviewer 2 ·

Basic reporting

The manuscript is well-written with clear, professional use of the English language. The authors provide a thorough and well-contextualized background, establishing the relevance of the study within its field. The overall structure aligns with the standards of PeerJ and falls within the journal's scope. Importantly, the references are appropriately cited throughout the manuscript. Given the relative scarcity of recent comprehensive reviews in this area, this work makes a timely and valuable contribution by synthesizing a wide range of important studies, thereby offering a solid foundation for future research.

Experimental design

The methodology employed in this study is clearly described and appropriately detailed. The manuscript aligns well with the aims and scope of PeerJ, and the authors have presented their findings in a coherent and logically structured manner. The literature has been thoroughly and accurately cited, contributing to the overall clarity and rigor of the work.

Validity of the findings

The conclusions of the study are clearly articulated and effectively address the original research question. The authors provide a comprehensive and well-structured explanation of the various diseases associated with the NLRP3 inflammasome, detailing the underlying mechanisms in a clear and concise manner. Notably, the manuscript also discusses additional conditions, such as myocardial infarction, where NLRP3 activation plays a significant role. Furthermore, the review thoughtfully summarizes ongoing research into therapeutic strategies targeting inflammasome inhibition and offers insightful suggestions for future improvement and development in this area.

Additional comments

The manuscript provides an insightful exploration of the role of the NLRP3 inflammasome in gastrointestinal diseases such as ulcerative colitis, Crohn’s disease, pancreatitis, and NAFLD. The authors are to be commended for effectively highlighting current research developments and potential avenues for future investigation. However, I note that the content between lines 73 and 76 reiterates points already presented in the abstract, resulting in some redundancy that could be refined for conciseness. Additionally, the description of the NLRP3 inflammasome as "novel" in line 105 may be reconsidered, given that it was first identified in 2002. Despite these minor concerns, the study represents a valuable contribution to the field.

Reviewer 3 ·

Basic reporting

In this manuscript, Liu and colleagues discuss the general characteristics of NLRP3 inflammasome, including its components and activation mechanisms. In addition, the authors explain how NLRP3 inflammasome is associated with the pathogenesis and progression of gastrointestinal diseases such as ulcerative colitis, Crohn’s disease, pancreatitis, and non-alcoholic fatty liver disease. While the review provides informative insights into the disease relevance of NLRP3 inflammasome, I made some suggestions to improve the manuscript before it can be considered for publication.

In the abstract, the authors clearly describe the topics that will be addressed in the paper. However, for the readers who are not familiar with the inflammasome, it would be beneficial to begin the abstract with a brief general background on the inflammasome, for example, its definition, components, or immunological relevance.

Experimental design

Line 69 & Lines 106-107: The phrases are somewhat vague in clearly conveying the role of inflammasomes, bridging between immune responses/immunity and inflammation. Inflammation is part of immunity or immune responses, so this could sound redundant or conceptually unclear. It would be helpful to specify the processes that the inflammasomes can bridge.

Lines 221-223: The interpretation of the cited paper’s results does not appear to be appropriate. Please review the original paper’s scope and findings carefully, or consider citing a more relevant reference to support the authors’ explanation.

The authors mentioned that clinically available NLRP3 inflammasome inhibitors for gastrointestinal diseases are still limited. It would be helpful to provide additional context or explanation, such as specificity, potential safety concerns, or difficulties in development.

Validity of the findings

The authors present a well-supported argument that aligns with the topics outlined in the introduction.

Additional comments

Line 204: The word is unnecessarily capitalized in the sentence. Please consider revising it.

The manuscript would benefit from professional editing to correct typos and grammar issues and to improve overall clarity.

---

## Round 0.2 · accepted · Accept

All critiques of the reviewers were adequately addressed, and the manuscript was revised accordingly. The manuscript is acceptable now.